# Advancements in Artificial Intelligence for Fetal Neurosonography: A Comprehensive Review

**DOI:** 10.3390/jcm13185626

**Published:** 2024-09-22

**Authors:** Jan Weichert, Jann Lennard Scharf

**Affiliations:** 1Division of Prenatal Medicine, Department of Gynecology and Obstetrics, University Hospital of Schleswig-Holstein, Ratzeburger Allee 160, 23538 Luebeck, Germany; jannlennard.scharf@uksh.de; 2Elbe Center of Prenatal Medicine and Human Genetics, Willy-Brandt-Str. 1, 20457 Hamburg, Germany

**Keywords:** fetal, ultrasound, prenatal, artificial intelligence, neurosonography, machine learning, convolutional neural networks

## Abstract

The detailed sonographic assessment of the fetal neuroanatomy plays a crucial role in prenatal diagnosis, providing valuable insights into timely, well-coordinated fetal brain development and detecting even subtle anomalies that may impact neurodevelopmental outcomes. With recent advancements in artificial intelligence (AI) in general and medical imaging in particular, there has been growing interest in leveraging AI techniques to enhance the accuracy, efficiency, and clinical utility of fetal neurosonography. The paramount objective of this focusing review is to discuss the latest developments in AI applications in this field, focusing on image analysis, the automation of measurements, prediction models of neurodevelopmental outcomes, visualization techniques, and their integration into clinical routine.

## 1. Introduction

The assessment of the anatomic integrity of the fetal central nervous system (CNS) is one of the most challenging tasks during a prenatal sonographic work-up, as the brain’s development and maturation constitute complex and well-orchestrated processes occurring at various embryonic and fetal stages. To preclude diagnostic errors, national and international guidelines explicitly pay attention to the fact that the appearance of the brain undergoes profound changes throughout gestation. Although brain anomalies are among of the most common fetal malformations, with an estimated prevalence of 9.8–14 per 10,000 live births [1,2], their in utero perception fundamentally requires a familiarity with sonographic brain anatomy and artifacts and a designated vigilance for the necessity of a subsequent targeted multiplanar assessment of the entire fetal CNS (neurosonography) [3,4]. In general, the efficacy of ultrasound (US) screenings largely hinges on the operator’s skill in navigating to and reproducing standard imaging planes and this, in turn, strongly relates to the gestational age (GA) at examination. In this context, it could be noted that, in the recent past, the majority of severe congenital brain anomalies have been readily identified prenatally by applying a systematic, protocol-based US survey [5,6]. Nevertheless, the detection rates of fetal brain lesions in an unselected population remain somewhat unsatisfactory and more subtle changes might escape an early diagnosis. In part, this might be explained by the fact that even though advanced technologies such as three-dimensional US (3DUS) undoubtedly have the potential to contribute to an improved detailed CNS evaluation, there is still little consensus as to the ideal method for volume acquisition, the settings, and the analysis of the volume and an overall lack in the standardization of volumetric assessments [7]. On the other hand, DiMascio et al. stated in their systematic review that fetal brain charts suffer substantially from poor methodologies and are at high risk of biases, especially when focusing on relevant neurosonographic issues [8]. In addition, another publication demonstrated that the fetal cortical brain’s development in fetuses conceived by assisted reproductive technology seems to be different from those conceived spontaneously, as expressed by a reduced sulci depth [9]. This underpins the complexity of an all-encompassing thorough assessment of the fetal brain. Beyond any doubt, prenatal US is capable of providing precise information regarding fetal anatomical integrity and the severity of abnormal conditions derived from high-quality images with increased diagnostic accuracy and reliability. The transabdominal route remains the technique of choice for a comprehensive anatomic evaluation of specific organs like the fetal brain. This clearly demonstrates that the currently available data source of images has to deal with a combination of maternal, fetal, technical, environmental, and acoustic factors hampering image clarity and data acquisition to eventually establish precise antenatal diagnoses.

Current research approaches regarding the clinical applicability of artificial intelligence (AI)-assisted methods in the context of fetal neurosonography (beyond the first trimester) are heterogeneous and, with a few exceptions, software solutions that are of use in clinical routine are rather rare. However, several promising research topics in this field have emerged. These mainly include (among others) the optimized (automated) acquisition of standard 2D planes with the correct orientation and localization within a 3DUS volume, a simplified workflow, the automated recognition of crucial CNS and bony structures (as landmarks) and the subsequent detection of anomalies, the evaluation of image quality and the assessment of GA by evaluating neurodevelopmental maturation [10,11,12,13,14].

This review aims to summarize our current knowledge about the potential diagnostic targets for AI algorithms in the assessment of the fetal brain in a clinical context and highlights why AI applications are increasingly being integrated into prenatal US interrogations and their practical added value.

## 2. AI in Prenatal Diagnosis

In the very recent past, we witnessed a tidal wave of artificial intelligence and its computational applications in healthcare in general and in medical image analyses in particular. In 2022, Dhombres et al. published a systematic review regarding the actual contributions of AI reported in obstetrics and gynecology (OB/GYN) journals. In detail, most articles covered method/algorithm development (53%, 35/66), hypothesis generation (42%, 28/66), or software development (3%, 2/66). Validation was performed on one dataset (86%, 57/66), while no external validation was reported [15].

Machine learning (ML) is a powerful set of computational tools that learn from large (structured) datasets and train models on descriptive patterns, which subsequently apply the knowledge acquired to solve the same task in new situations. Although ML algorithms are presently being widely deployed in medicine, expanding diagnostic and clinical tools to augment iterative, time-consuming, and resource-intensive processes and streamline workflows, considerable human supervision is needed. AI models that use deep learning architectures (DL; a subdomain of ML), which predominantly leverage large-scale neural networks mimicking silicon circuit synapses, tend to outperform traditional machine learning methods in complex tasks and constitute the most suitable methodology for image analysis. Based on a detailed scoping review dealing with the most-cited papers using DL in the literature from 2015 to 2021, the number of surveyed publications on segmentation, detection, classification, registration, and characterization tasks comprised 30, 20, 30, 10, and 10 percent, respectively [16]. It is of note that the quality of obstetric US screening images is crucial for clinical downstream tasks, including the assessment of fetal growth and development, in utero compromising, the prediction of preterm birth, and the detection of fetal anomalies. It is now widely recognized, by leading US equipment manufacturers and most of the experts in this field, that there are clear benefits to utilizing AI technologies for US imaging in prenatal diagnostics. A multitude of convolutional neural network (CNN)-based AI applications in US imaging have showcased that AI models can achieve a comparable performance to clinicians in obtaining the appropriate diagnostic image planes, applying appropriate fetal biometric measurements, and accurately assessing abnormal fetal conditions [10,11,12,13,17,18].

As accurate head measurements are of crucial importance in prenatal and obstetrical ultrasound surveillance, a plethora of automatic methods for fetal head analyses have been proposed. Most studies have focused preferentially on the size and shape of the bony skull—excluding its internal structures—while solely applying head detection methods (e.g., object (skull) detection using bounding boxes, segmentation methods ± ellipse fitting, and edge-based and contour-based methods). Torres et al. published an excellent comprehensive state-of-the-art review tabulating more than 100 published papers on computational methods for fetal head, brain, and standard plane analyses using US images [13]. Moreover, their survey also summarized the image enhancement protocols of US images, including methods that find the fetal head aligned to a coordinate system, compounding approaches, and US and multimodal registration methods. The authors provided an exhaustive analysis of each method based on its clinical application and theoretical approach and in their concluding remarks they stated—despite the fact that a multitude of distinct image processing methods have been developed in the recent past (which mainly comprise deep learning approaches)—that there is a need for new architectures to boost the performance of these methods. A strong database is seen as an indispensable prerequisite, which reinforces the need for public US benchmarks and for the development of approaches that deal with limited data (e.g., transfer learning approaches). On the other hand, more effort should be made in AI research to develop methods to segment the head in 3D images, as well as methods that can reliably detect abnormalities or (even subtle) lesions within the fetal brain. Accordingly, Ramirez Zegarra and Ghi suggested an ideal AI setting which clearly addresses the urgent need for more multitasking DL models that are trained for the detection of fetal standard planes, the identification of fetal anatomical structures, and the performance of automatic measurements that in turn will consequently be able to generate alarm messages in the event of malformations [19].

In fact, in the last decade, several AI-related scientific studies have been conducted to improve the quality of prenatal diagnoses by focusing on three major issues: (I) the detection of anomalies, fetal measurements, scanning planes, and the heartbeat; (II) the segmentation of fetal anatomic structures in still US frames and videos; and (III) the classification of standard fetal diagnostic planes, congenital anomalies, biometric measures, and fetal facial expressions [20].

Various researchers were able to develop algorithms that were able to reproducibly quantify biometric parameters with high accuracy. Some of them will be discussed critically below. On the other hand, quite a number of AI models were trained with inadequate and/or insufficiently labeled samples, which led to overfitting problems and performance degradation [14]. ‘All models are wrong, but some are useful’, is an aphorism on the subject of statistics coined by George E.P. Box more than 50 years ago, and one that describes the general dilemma in the targeted application of computational modeling approaches and AI solutions (and not only in the past) [21].

## 3. AI in Fetal Neurosonography

By adopting experience from the use of automated techniques in fetal cardiac assessments, further refinements in AI algorithms or the development of anomaly specific learning algorithms could help achieve more granular detections of unique CNS lesions [22]. This has the potential to risk stratifying certain fetal populations. But, as a matter of fact, it has to be acknowledged that algorithms developed for fetal imaging recognition require a larger database compared to other AI algorithms, due to the similar appearance of different US planes [19]. However, it should be noted that the currently available and clinically approved approaches for the extensive processing of three-dimensional datasets of the fetal CNS only sparsely exploit the diagnostic potential of volume US. In fact, it is currently not feasible to perform both simple and more complex tasks simultaneously, such as an assessment of the total brain volume, rendering the brain surface and slicing and displaying all diagnostic sectional planes (in accordance with the ISUOG guidelines), in addition to a multiplanar image display using the same 3D volume (in a vendor-independent manner). This applies to both conventional tools and AI frameworks and does not appear to be readily explainable in the light of the existing scientific literature, with its generally highly complex AI pipelines. In this regard, developers and engineers, on the one hand, and clinicians, on the other, should work together more intensively to find relevant integrative volume-based solutions for clinical routine as quickly as possible.

In 2023, an international research group developed a normative digital atlas of fetal brain maturation based on a prospective international cohort (INTERGROWTH-21st Project) and using more than 2500 serially acquired 3D fetal brain volumes [23]. In the preparation of this fully functional digital brain atlas, the authors proposed an end-to-end, multi-task CNN that both extracts and aligns the fetal brain from original 3D US scans with a high degree of accuracy and reliability (Brain Extraction and Alignment Network; BEAN) [24,25,26]. These steps were necessary (as in most neuroimage analysis pipelines) to enhance the visibility of the brain structures within the 3D US templates and significantly reduce the amount of extra-cranial volume information processed and, lastly, overcome the positional variation of the brain inside the scan volume. From the authors’ perspective, there is no doubt that the introduction of computerized human body atlases either based on US or MRI image data (as published earlier by Gholinpour et al. [27]) will contribute to our understanding of fetal developmental processes in general and brain maturation in particular by providing rich contextual information of our inherently 3D (CNS) anatomy.

The clinical applicability of semiautomatic volumetric approaches, in terms of a detailed reconstruction of the diagnostic planes of the fetal brain, has been validated in previous studies [28,29,30]. Very recently, a 3D UNet-based network for the 3D segmentation of the entire CNS, using intelligent navigation to locate CNS planes within the 3D volume, was introduced (fully automated 5DCNS+™). While applying this tool, our group was able to show that CNS volume datasets (whose acquisition was from an axial transthalamic plane) could readily be reconstructed into a nine-view template in less than 12 s on average, facilitating the generation of a complete neurosonogram with high accuracy, efficiency, and reduced operator-dependency, confirming previous findings.

Lu et al. reported on an automated software (Smart ICV™) that was able to calculate the entire fetal brain volume retrieved from 3DUS volume data. This novel technique showed a high intra- and inter-observer intra-class coefficient (0.996 and 0.995, respectively) and high degree of reliability compared to a manual approach using Virtual Organ Computer-aided AnaLysis (VOCAL™) [31]. An overview of the current AI-driven algorithms with either a clinical or pre-clinical context is given in Table 1.

DL algorithms have become the methodology of choice for imaging analyses [16,18,32,33]. DL models are capable of overcoming US-image-related challenges including inhomogeneities, (shadowing) artifacts, poor contrast, intra- and inter-clinician data acquisition, and measurement variability. Fiorentino and co-workers categorized published work in the field of fetal US image analysis that used a plethora of different DL algorithms. Their review surveyed more than 150 research papers to elaborate the most investigated tasks addressed using DL in this field [18]. The authors could demonstrate that fetal standard plane (SP) detection (19.6%) and fetal biometry estimations (20.9%) were among the most prevalent tasks. The fetal CNS and heart were the most explored structures in standard plane detection, while the fetal head circumference was the most frequently investigated measurement in biometry estimations. In 49% of papers, researchers trained DL pipelines for anatomical structure analyses. The most studied anatomical structures were the heart and brain, contributing to 26.7% and 20.0% of the surveyed papers, respectively. In case of the latter, the analysis is performed on both 2D and, more recently, 3D images to assess the brain’s development and localization, structure segmentation, and GA estimation. The challenges to be addressed regarding AI in image analyses, in general, comprise the limited availability of (multi-expert) image annotation; the limited robustness of DL algorithms due to the lack of large training datasets (interestingly, only a minority of DL studies use data from routine clinical care); the inconsistent use of both performance metrics and testing datasets, hampering a fair comparison between different algorithms; and the scarcity of research proposing semi-, weakly, or self-supervised approaches [10,18,32].

**Table 1 jcm-13-05626-t001:** Neurosonographic studies related to artificial intelligence.

Reference, Year	Country	GA (wks)	Study Size (n) *	Data Source	Type of Method	Purpose /Target	Task	Description of AI	Clinical Value ***
Rizzo et al., 2016 [34]	I	21 (mean)	120	3D	n. s.	SFHP (axial) biometry	automated recognition of axial planes from 3D volumes	5D CNS software	++
Rizzo et al., 2016 [35] **	I	18–24	183	3D	n. s.	SFHP (axial/ sagittal/coronal) biometry	evaluation of efficacy in reconstructing CNS planes in healthy and abnormal fetuses	5D CNS+ software	+++
Ambroise-Grandjean et al., 2018 [36]	F	17–30	30	3D	n. s.	SFHP (axial) biometry (TT, TC)	automated identification of axial from 3DUS and measurement BPD and HC	SmartPlanes CNS	++
Welp et al., 2020 [30] **	D	15–36	1110	3D	n. s.	SFHP (axial/ sagittal/coronal) biometry	validating of a volumetric approach for the detailed assessment of the fetal brain	5D CNS+ software	+++
Pluym et al., 2021 [37]	USA	18–22	143	3D	n. s.	SFHP (axial) biometry	evaluation of accuracy of automated 3DUS for fetal intracranial measurements	SonoCNS software	++
Welp et al., 2022 [29] **	D	16–35	91	3D	n. s.	SFHP/anomalies biometry	evaluation of accuracy and reliability of a volumetric approach in abnormal CNSs	5D CNS+ software	+++
Gembicki et al., 2023 [28] **	D	18–36	129	3D	n. s.	SFHP (axial/ sagittal/coronal) biometry	evaluation of accuracy and efficacy of AI-assisted biometric measurements of the fetal CNS	5D CNS+ software, SonoCNS software	++
Han et al., 2024 [38]	CHN	18–42	642	2D	DL	Biometry (incl. HC, BPD, FOD, CER, CM, Vp)	automated measurement and quality assessment of nine biometric parameters	CUPID software	++
Yaqub et al., 2012 [39]	UK	19–24	30	3D	ML	multi-structure detection	localization of four local brain structures in 3D US images	Random Forest Classifier	++
Cuingnet et al., 2013 [40]	UK	19–24	78 volumes	3D	ML	SFHP	fully automatic method to detect and align fetal heads in 3DUS	Random Forest Classifier, Template deformation	++
Sofka et al., 2014 [41]	CZ	16–35	2089 volumes	3D	ML	SFHP	automatic detection and measurement of structures in CNS volumes	Integrated Detection Network (IDN)/FNN	+
Namburete et al., 2015 [42]	UK	18–34	187	3D	ML	sulcation/gyration	GA prediction	Regression Forest Classifier	++
Yaqub et al., 2015 [43]	UK	19–24	40	3D	ML	SFHP	extraction and categorization of unlabeled fetal US images	Random Forest Classifier	+
Baumgartner et al., 2016 [44]	UK	18–22	201	2D	DL	SFHP (TT, TC)	retrieval of standard planes, creation of saliency maps to extract bounding boxes of CNS anatomy	CNN	+++
Sridar et al., 2016 [45]	IND	18–20	85	2D	DL	structure detection	image classification and structure localization in US images	CNN	+
Yaqub et al., 2017 [46]	UK	19–24	40	3D	DL	SFHP, CNS anomalies	localization of CNS, structure detection, pattern learning	Random Forest Classifier	+
Qu et al., 2017 [47]	CHN	16–34	155	2D	DL	SFHP	automated recognition of six standard CNS planes	CNN, Domain Transfer Learning	++
Namburete et al., 2018 [25]	UK	18–34	739 images	2D/3D	DL	structure detection	3D brain localization, structural segmentation and alignment	multi-task CNN	++
Huang et al., 2018 [48]	CHN	20–29	285	3D	DL	multi-structure detection	detection of CNS structures in 3DUS and measurements of CER/CM	VP-Net	++
Huang et al., 2018 [49]	UK	20–30	339 images	2D	DL	structure detection (CC/CP)	standardize intracranial anatomy and measurements	Region descriptor, Boosting classifier	++
van den Heuvel et al., 2018 [50]	NL	10–40	1334 images	2D	ML	biometry (HC)	automated measurement of fetal head circumference	Random Forest Classifier Hough transform	+
Dou et al., 2019 [51]	CHN	19–31	430 volumes	3D	ML	SFHP/structure detection	automated localization of fetal brain standard planes in 3DUS	Reinforcement learning	++
Sahli et al., 2019 [52]	TUN	n/a	86	2D	ML	SFHP	automated extraction of biometric measurements and classification of normal/abnormal	SVM Classifier	++
Alansary et al., 2019 [53]	UK	n/a	72	3D	ML/DL	SFHP/structure detection	localization of target landmarks in medical scans	Reinforcement learning deep Q-Net	+
Lin et al., 2019 [54]	CHN	14–28	1771 images	2D	DL	SFHP/structure detection	automated localization of six landmarks and quality assessments	MF R-CNN	+
Bastiaansen et al., 2020 [55]	NL	1st trimester	30	2D/3D	DL	SFHP (TT)	fully automated spatial alignment and segmentation of embryonic brains in 3D US	CNN	+
Xu et al., 2020 [56]	CHN	2nd/3rd trimester	3000 images	2D	DL	SFHP	simulation of realistic 3rd- from 2nd-trimester images	Cycle-GAN	++
Ramos et al., 2020 [57]	MEX	n/a	78 images	2D	DL	SFHP biometry (TC) GA prediction	detection and localization of cerebellum in US images, biometry for GA prediction	YOLO	+
Maraci et al., 2020 [58]	UK	2nd trim	8736 images	2D	DL	biometry (TC) GA prediction	estimation of GA through automatic detection and measurement of the TCD	CNN	+
Chen et al., 2020 [59]	CHN	n/a	2900 images	2D	DL	SFHP biometry (TV)	demonstrate the superior performance of DL pipeline over manual measurements	Mask R-CNN ResNet50	+
Xie et al., 2020 [60]	CHN	18–32	92,748	2D	DL	SFHP (TV, TC) CNS anomalies	image classification as normal or abnormal, segmentation of craniocerebral regions	U-Net VGG-Net	++
Xie et al., 2020 [61]	CHN	22–26	12,780	2D	DL	SFHP, CNS anomalies	binary image classification as normal or abnormal in standard axial planes	CNN	++
Zeng et al., 2021 [62]	CHN	n/a	1354 images	2D	DL	biometry	image segmentation for automatic HC biometry	DAG V-Net	+
Burgos Artizzu et al., 2021 [63]	ESP	16–42	12,400 images (6041 CNS)	2D	DL/ML	SFHP	evaluation of the maturity of current DL classifications tested in a real clinical environment	19 different CNNs MC Boosting algorithm HOG classifier	++
Gofer et al., 2021 [64]	IL	12–14	80 images	2D	ML	SFHP/structure detection (CP)	classification of 1st trimester CNS US images and earlier diagnosis of fetal brain abnormalities	Statistical Region Merging Trainable Weka Segmentation	+
Skelton et al., 2021 [65]	UK	20–32	48	2D/3D	DL	SFHP	assessment of image quality of CNS planes automatically extracted from 3D volumes	Iterative Transformation Network (ITN)	++
Fiorentino et al., 2021 [66]	I	10–40	1334 images	2D	DL	biometry (HC)	head localization and centering	multi-task CNN	++
Yeung et al., 2021 [67]	UK	18–22	65 volumes	2D/3D	DL	SFHP/structure detection	mapping 2D US images into 3D space with minimal annotation	CNN	
Montero et al., 2021 [68]	ESP	18–40	8747 images	2D	DL	SFHP	generation of synthetic US images via GANs and improvement of SFHP classification	Style-GAN	++
Moccia et al., 2021 [69]	I	10–40	1334 images	2D	DL	biometry (HC)	fully automated method for HC delineation	Mask-R2CNN	+
Wyburd et al., 2021 [70]	UK	19–30	811 images	3D	DL	structure detection/ GA prediction	automated method to predict GA by cortical development	VGG-Net ResNet-18 ResNet-10	++
Shu et al., 2022 [71]	CHN	18–26	959 images	2D	DL	SFHP (TC)	automated segmentation of the cerebellum, comparison with other algorithms	ECAU-Net	+
Hesse et al., 2022 [72]	UK	18–26	278 images	3D	DL	structure detection	automated segmentation of four CNS landmarks	CNN	+++
Di Vece et al., 2022 [73]	UK	20–25	6 volumes	2D	DL	SFHP/structure detection	estimation of the 6D pose of arbitrarily oriented US planes	ResNet-18	++
Lin et al., 2022 [74]	CHN	18–40	16,297/166	2D	DL	structure detection	detection of different patterns of CNS anomalies in standard planes	PAICS YOLOv3	+++
Sreelakshmy et al., 2022 [75] ‡	IND	18–20	740 images	2D	DL	biometry (TC)	cerebellum segmentation from fetal brain images	ResU-Net	-
Yu et al., 2022 [56]	CHN	n/a	3200 images	2D/3D	DL	SFHP	automated generation of coronal and sagittal SPs from axial planes derived from 3DVol	RL-Net	++
Alzubaidi et al., 2022 [76]	QTAR	18–40	551	2D	DL	biometry (HC)	GA and EFW prediction based on fetal head images	CNN, Ensemble Transfer Learning	++
Coronado-Gutiérrez et al., 2023 [77]	ESP	18–24	12,400 images	2D	DL	SFHP, multi-structure delineation	automated measurement of brain structures	DeepLab CNNs	++
Ghabri et al., 2023 [20]	TN	n/a	896	2D	DL	SFHP	classify fetal planes/accurate fetal organ classification	CNN: DenseNet169	++
Lin et al., 2023 [78]	CHN	n/a	558 (709 (images/videos)	2D	DL	SFHP	improved detection efficacy of fetal intracranial malformations	PAICS YOLO	+++
Rauf et al., 2023 [79]	PK	n.s.	n.s.	2D	DL	SFHP	Bayesian optimization for the classification of brain and common maternal fetal ultrasound planes	Bottleneck residual CNN	+
Alzubaidi et al., 2023 [80]	QTAR	18–40	3832 images	2D	DL	SFHP	evaluation of a large-scale annotation dataset for head biometry in US images	multi-task CNN	+
Alzubaidi et al., 2024 [81]	QTAR	18–40	3832 images (20,692 images)	2D	DL	biometry	advanced segmentation techniques for head biometrics in US imagery	FetSAM Prompt-based Learning	+
Di Vece et al., 2024 [82]	UK	20–25	6 volumes	2D/3D	DL	SFHP (TV)	detection and segmentation of the brain; plane pose regression; measurement of proximity to target SP	ResNet-18	++
Yeung et al., 2024 [83]	UK	19–21	128,256 images	2D	DL	SFHP	reconstruction of brain volumes from freehand 2D US sequences	PlaneInVol ImplicitVol	++
Dubey et al., 2024 [84]	IND	10–40	1334 images	2D	DL	biometry (HC)	automated head segmentation and HC measurement	DR-ASPnet, Robust Ellipse Fitting	++

Clinically validated (and commercially available) software in gray shaded rows. Abbreviations: 2D, two dimensional; 3D, three dimensional; BPD, biparietal diameter; CER, cerebellum; CNN, convolutional neural network; CNS, central nervous system; CP, choroid plexus; CSP, cavum septum pellucidum; DL, deep learning; FOD, fronto-occipital diameter; GA, gestational age; GAN, generative adversarial network; HC, head circumference; LV, lateral ventricles; n/a, not applicable; n.s., not specified; PAICS, prenatal ultrasound diagnosis artificial intelligence conduct system; ResNet, residual neural network; SFHP, standard fetal head plane; SVM, support vector machine; TC, transcerebellar plane; TV, transventricular plane; TT, transthalamic plane; US, ultrasound; Vp, width of the posterior horn of the lateral ventricle; YOLO, You Only Look Once algorithm; * if not otherwise specified: number of patients; ** fully automated AI-driven software update has been released; *** potential clinical impact; ‡ withdrawn article.

### 3.1. AI in GA Prediction

Reliable methods for accurate GA estimations in the second and third trimester of pregnancy remain an unsolved challenge in obstetrics. This might be due to late booking, infrequent access to prenatal care, the unavailability of early US examinations, and other reasons [63]. Namburete et al. introduced a model which was able to characterize neuroanatomical appearance, both spatially and temporally, while identifying relevant brain regions, such as the Sylvian fissure and the cingulate and callosal sulci, as important image regions in the GA discrimination task [42]. The authors additionally extended this to clinically relevant metadata like the head circumference’s canonical feature set (e.g., Haar-like features) to capture structural changes within the fetal brain. The algorithm improved the confidence of age predictions provided by the clinical HC method by ±0.64 days and ±4.57 days in the second and third trimesters, respectively. A similar approach estimates GA from standard transthalamic axial plane images using a supervised DL model (quantusGA) that automatically detects the position and orientation of the fetal brain by detecting the skull and five internal key points (it is necessary to crop and rotate the brain, resulting in a horizontally aligned brain image). The model then extracts textural and size information from the brain pixels and uses this information to generate an estimate of the respective GA, with a similar or even lower error compared to fetal biometric parameters, especially in the third trimester [63]. AI models are capable of the estimation of GA with an accuracy comparable to that of trained sonographers conducting a standard fetal biometry (e.g., the fetal head), as the results of a recent study suggest. The authors trained a DL algorithm to estimate GA from blind US sweeps and showed that the model’s performance appears to extend to blind sweeps collected by untrained providers in low-resource settings [84]. Similar results were demonstrated by two groups, where ML-based algorithms outperformed current ultrasound-based clinical biometry in GA prediction, with a mean absolute error of 3.0 and 4.3 days [85] or 1.51 days (using an ensemble model of both unlabeled images and video data) in second- and third-trimester fetuses [86].

### 3.2. AI Used for Augmenting Fetal Pose Estimations and CNS Anomaly Assessments

In the recent past, CNNs and other deep learning architectures were trained to recognize and predict fetal poses from imaging data [25,66,87,88]. In contrast to already established methods, which were mainly designed for standard plane identification, assuming that a good US image quality had already been achieved with the fetus in a proper position, and, therefore, only used to assist in prenatal image analyses, a study group from the UK recently emphasized the utility of recognizing the probe’s proximity to diagnostic CNS planes, facilitating earlier and more precise adjustments during 2D US scanning. This semi-supervised segmentation and classification model used an 18-layer residual CNN (ResNet-18) that was trained on both labeled standard planes and unlabeled 3D US volume slices to filter out frames lacking the brain and to generate masks for those containing it, enhancing the relevance of plane pose regression in a clinical setting [81]. In a previous study, the authors applied a similar 18-layer residual CNN as the backbone for feature extraction (with pre-trained ImageNet weights) and 6D pose prediction (which refers to the task of determining the sixth degree-of-freedom pose of an object in 3D space) of arbitrarily oriented planes, slicing the fetal brain’s US volume without the need for ground truth data in real time or 3D volume scans of the fetus beforehand [72].

Yeung and colleagues proposed an algorithm for the more general task of predicting the location of any arbitrary 2D US plane of the fetal brain in a pre-defined 3D space [66]. In their work, they demonstrated that, based on extensive data augmentation and complementary information from training volumes acquired at different orientations, the prediction made by a novel CNN model was generalizable to real 2D US acquisitions and videos, despite the model having only been trained with artificially sampled 2D slices. Considering that 3D volumes provide more effective spatial information and exhibit higher degrees of freedom (DoF), increased variations in fetal poses make the proper education of these algorithms challenging. In fact, 6D fetal pose estimation refers to the process of determining the six degrees of freedom (6DoF) position including three translational (position) and three rotational (orientation) parameters, allowing for a comprehensive understanding of the fetus’s spatial position in a coordinate system and movement in utero. The study conducted by Chen and co-workers dealt with a similar topic—fetal pose estimations in 3D US. The authors introduced a novel 3D fetal pose estimation framework (FetusMapV2) which was able to identify a set of 22 anatomical landmarks for first- and early second-trimester fetuses, and their specific connections, to provide a comprehensive and systematic representation of the fetal pose in 3D space and to overcome challenging issues such as poor image quality, limited GPU memory for tackling high-dimensional data, symmetrical or ambiguous anatomical structures, and considerable variations in fetal poses [87].

Xu and co-workers trained a DL algorithm (a cycle-consistent adversarial network (Cycle-GAN)) to simulate realistic fetal neurosonography images and specifically to generate third-trimester US images from second-trimester images that were qualitatively evaluated by experienced sonographers [56]. The vast majority (84.2%) of the simulated third-trimester images could not be distinguished from real third-trimester images in this study. These generative adversarial networks (GANs), first introduced by Goodfellow et al. in 2014, are algorithmic architectures that use two neural networks, which compete against each other and learn to generate new, synthetic instances of data, with a probabilistic model, that can pass for real data/images, augmenting existing datasets for training DL models [89,90]. Generative approaches can better handle missing data in multi-modal datasets by generating the missing image information and preserving the sample size, thereby boosting downstream classification performances [91,92]. GANs might also assist in analyzing abnormal fetal anatomical structures (e.g., CNS anomalies) while also considering the corresponding GA information (there are a wide range of physiological changes among trimesters leading to marked inter- and intra-organ variability) [18]. A recent research paper introduced a state-of-the-art framework (FetalBrainAwareNet) that leverages an image-to-image translation algorithm and utilizes class activation maps (CAMs) as prior information in its conditional adversarial training process, making it capable of producing more realistic synthetic images, resulting, according to the authors, in a greater clinical relevance than similar experimental approaches [56,67,93,94]. The uniqueness of this approach was the incorporation of anatomy-aware regularization terms—one ensuring the generation of elliptical fetal skulls, while another was crucial for refining and distinctly differentiating key anatomical landmarks (e.g., cerebellum, thalami, cavum septi pellucida, lateral ventricles)—in each particular fetal head standard plane (FHSP).

In US imaging, the presence of speckle noise degrades the signal-to-noise of US images; traditional image denoising algorithms often fail to fully reduce speckle noise and retain the image’s features. A recently proposed GAN based on U-Net with residual dense connectivity (GAN-RW) achieved the most advanced despeckling performance on US images (e.g., of the fetal head) in terms of its peak signal-to-noise ratio (PSNR), structural similarity (SSIM), and subjective visual effect [95]. Yeung et al. proposed a novel framework (ImplicitVol), a sensor-free approach to reconstructing 3D US volumes from a sparse set of 2D images with deep implicit representation. The authors stated that their algorithm outperformed conventional approaches in terms of the image quality of the reconstructed template, as well as the refinement of its spatial 3D localization, which underscored its additional potential in slice-to-volume registration [82]. The latter refers to the vital technique in medical imaging that transforms 2D slices into a cohesive 3D volume, thereby enhancing our ability to visualize and analyze complex anatomical structures, leading to optimized diagnostic (and therapeutic) outcomes. The same group introduced a multilayer perceptron network (RapidVol) to speed up slice-to-volume ultrasound reconstruction following a tri-planar decomposition of original 3D brain volumes and were able to demonstrate a threefold quicker and 46% more accurate complete 3D reconstruction of the fetal brain (collected as part of the INTERGROWTH-21st study) compared to the aforementioned implicit approach [96].

Lin et al. developed a real-time artificial intelligence-aided image recognition system based on the YOLO (You Only Look Once) algorithm (prenatal ultrasound diagnosis artificial intelligence conduct system; PAICS) which was capable of detecting a set of fetal intracranial malformations. The algorithm was trained on 44,000 images and 169 videos and achieved an excellent performance upon both internal and external validation, with an accuracy comparable to that of expert sonologists [73]. The same group conducted a randomized control trial that assessed the efficacy of a deep learning system (PAICS) in assisting in fetal intracranial malformation detection. More than 700 images/videos were interactively assessed by 36 operators with different levels of expertise. With the use of PAICS (prior or after individual interpretation) an increase in the detection rates of fetal intracranial malformations from neurosonographic data could be noticed [77].

### 3.3. Other Current AI Applications Related to Fetal Neurosonography

The recent and rapidly emerging subfield of AI that concerns the interaction between computers and human language is known as natural language processing (NLP). The launch of the chatbot ChatGPT-3, a large language model (LLM), in 2022, which is based on an NLP model known as the Generative Pretrained Transformer (GPT), has generated a wide range of possible applications for AI in healthcare [97,98,99]. Therefore, beyond the field of (often cited) scientific writing, identifying suitable areas of its application in obstetrics and gynecology, including fetal neurosonography, is obvious. It is crucial to understand that ChatGPT, which is trained on massive amounts of text data, mimics statistical patterns of human language and generates outputs based on probabilities, thus emulating the dynamics of human conversation [97,98,100,101]. Before addressing the potential applications of ChatGPT in the context of fetal neurosonography, two fundamental, capability-limiting aspects must be kept in mind and must never be ignored in the interpretation of the subsequent discussion: Although ChatGPT should increasingly become capable of generating meaning-semblant behavior, its technology currently lacks semantic understanding [102]. Be aware of hallucinations and fabricated facts [98]. Furthermore, its generated content suffers from an absence of verifiable references [99,100,101].

Most recently, the latest version of ChatGPT (GPT-4) has been evaluated for its ability to facilitate referrals for fetal echocardiography to improve the early detection of and outcomes related to congenital heart defects (CHDs) [103]. Kopylov et al. found moderate agreement between ChatGPT and experts. Comparing AI referrals to experts indicated an agreement of around 80% (*p* < 0.001). For minor CHD cases, the AI referral rate was 65% compared to 47% for experts. In future, AI could presumably support clinicians in this area.

A similar approach would be conceivable for fetal neurosonography, as would the implementation of language-based AI to support in summarizing the findings and optimizing the wording of complex medical reports, or even in the classification of various sonographic CNS abnormalities into corresponding disease entities with differential diagnoses. In summary, ChatGPT cannot be used independently of experts in the field of fetal neurosonography and certainly will not replace them [100,104]. We agree with other authors that it is unlikely that ChatGPT, even in improved versions, will ever be able to provide reliable data at the standard required by evidence-based medicine [100,105]. However, repetitive, time-consuming tasks and conclusions made in clinical routine will soon be left to this chatbot.

## 4. Perspectives

AI-based applications, on whose algorithms prenatal diagnostics will increasingly depend, are fundamentally changing the way clinicians use US. Even if the development of AI-based applications in obstetric US is still in its infancy and automation has not yet reached the required level of clinical application, sometime soon the use of AI in fetal neurosonography will exceed the capabilities of human experts, as in other fields of fetal US [11,14].

A recently published paper introduced a novel approach that parameterizes 3D volume data using a deep neural network, which jointly refines the 2D-to-3D registrations and generates a full 3D reconstruction based on only a set of non-sensor-tracked freehand 2D scans [82].

Unfortunately, the black box nature of most machine learning models remains unresolved, and many decisions intelligent systems make still lack interpretable explanations. Explainable AI (XAI) is considered to provide methods, equations, and tools that make the results generated by an AI algorithm comprehensible for the user. By providing visual and feature-based explanations, XAI enhances the transparency and trustworthiness of AI predictions and could thus pave the way for the initial uptake of an AI model into clinical routine [106]. In this regard, a recent study analyzed the performance of several CNNs trained on 12,400 images of fetal (CNS) plane detection, after input (image) enhancement, by adopting a Histogram Equalization and Fuzzy Logic-based contrast enhancement. The results achieved an accuracy between 83.4 and 90% (depending on the classifier analyzed) and were subsequently evaluated by applying the LIME (Local Interpretable Model-Agnostic Explanations) and GradCAM (Gradient-weighted Class Activation Mapping) algorithms to examine the decision-making process of the classifiers, providing explainability for their outputs [107]. These XAI models visually depict the region of the image contributing to a particular class, thereby justifying why the model predicted that class [108]. Very recently, Pegios and co-workers used iterative counterfactual explanations to generate realistic high-quality CNS standard planes from low-quality non-standard ones. Using their experimental approach (Diff-ICE), they demonstrated its superior value in the challenging task of fetal ultrasound quality assessments, as well as its potential for future applications [109]. To alleviate the risks of incomprehensibility and—more crucially—clinical irrelevance in forthcoming research, two publications proposed directive guidelines for transparent ML systems in medical image analyses (INTRPRT/Clinical XAI Guidelines) [110,111]. Interestingly, all sixteen commonly used heatmap XAI techniques evaluated by Jin et al. were found to be insufficient for clinical use due to their failure in the criteria of ‘truthfulness’ and ‘plausibility’ [111].

Acknowledging the recent achievements of AI in medical image analyses, Sendra-Balcells and co-workers addressed the paradox that is that the development of AI in rural areas in the world, like in Sub-Saharan Africa, is at its lowest level, while, on the other hand, current AI advancements include deep learning implementations in prenatal US diagnoses, which can facilitate improved antenatal screening. In this regard, they investigated the generalizability of fetal US deep learning models to low-resource imaging settings [112]. The authors pre-trained a DL framework for standard plane detection (e.g., the fetal brain) in centers with greater access to large clinical imaging datasets and subsequently applied this model to African settings. The results gained from transfer learning exemplify that domain adaptation might be a solution that supports prenatal care in low-income countries.

As a recent commentary given by Tonni and Grisolia correctly stated, we will inevitably have to face that the incorporation of AI solutions into the US apparatus will start to surge exponentially in the near future, producing beneficial effects not only in terms of diagnostic accuracy but also in the quality of fetal examination in its entirety, including the appropriate surveying of complex anatomical structures (e.g., the fetal CNS and heart), the reporting of these exams, and improving medical–legal issues for physicians involved in both fetal imaging and fetomaternal care [113].

## 5. Discussion

Our focusing review provides insights into both the current research topics and clinical applications of AI-based algorithms related to the field of fetal neurosonography and sheds light on how recent advancements in AI, and particularly cutting-edge technologies like GANs, segmentation-based approaches, XAI tools, and others, could further enhance the US image analysis of the fetal CNS. The strength of our review is the exclusive inclusion of publications addressing the state of the art of AI-driven methods for the US assessment of the fetal brain to enable clinicians to contextualize these applications in their clinical workup, illustrate potential pitfalls, and outline future avenues of fetal neurosonography to pursue. In fact, AI-driven models showcased how the accuracy, workflow efficiency, and interpretability of US imaging can be improved, which in turn might contribute to an earlier and more precise detection of fetal brain anomalies in utero. Prospectively, DL frameworks could be trained to detect structural abnormalities of the fetal brain, to label the type of malformation observed in diagnostic standard planes, and to generate alerts to prompt prenatal diagnoses. While 2D US remains the primary diagnostic tool for fetal neurosonography and (sequences of) 2D cross-sectional views of inherently 3D neuroanatomic structures are used to train AI algorithms, one must acknowledge a considerable loss of conceptional image information. The image data retrieved from 3D US with multiplanar reconstruction can complement conventional 2D US and overcome the limitations of the latter. Due to the well-described barriers (e.g., a lack of familiarity with volume postprocessing and skilled manual navigation) to the routine use of 3D US in prenatal diagnoses, recent advances in 3D imaging have focused on the implementation of intelligent algorithms for the automated extraction of data from 3D volume datasets. Several publications demonstrated the superior value of AI tools in facilitating a rapid, easier, and less operator-dependent 3D volume analysis of fetal CNS anatomy. Alternatively, it has been shown that 3D volumes can be effectively constructed from 2D scans by applying ML/DL approaches. The inherent advantages of DL-based slice-to-volume (or 2D/3D) registration techniques comprise a fully automated alignment and transfer of spatial information between subjects and imaging modalities and the ability to correct for motion and misaligned slices when reconstructing the volume of a certain modality. In this regard, an interesting approach for the fetal 6D pose estimation of cutting planes (relative to the fetal brain center) or the recently released normative brain atlas, which apply comparable AI pipelines to enhance the visibility of the fetal CNS in 3D US images, must be mentioned, as these underscore the tremendous educational potential of these algorithms. Our capability to make AI-augmented assessments of fetal brain maturation also allows for GA estimations using CNS image data with high accuracy, exemplifying its clinical value in low-resource obstetrical settings.

However, there are several limitations to this review article. The selection of the included studies was based on the authors’ subjective assessment of the methodology, diagnostic relevance, and potential of the innovative AI algorithms described therein to be integrated into (future) clinical workflows. Although we were able to address the advantages of particular promising AI approaches and their added value, an in-depth comparison was not possible due to the heterogeneity of these models and would go beyond the scope of this review.

## 6. Conclusions

AI has increasingly been accepted as a fundamental component of a multitude of healthcare applications, such as medical image analyses. In light of this inevitable and intriguing flooding of intelligent algorithms into modern US diagnostics, nothing less than the beginning of a new era of 5D ultrasound has been proclaimed. However, there are several challenges to AI’s deployment, particularly in fetal neurosonography, that must be solved: the need for large and diverse training datasets (2D/3D) in general; the difficulty of training accurate models for diagnosing evolving fetal brain abnormalities; the potential for algorithmic biases; the urgent need to address the troubling lack of transparency and interpretability of current AI algorithms to achieve their further translation into clinical diagnostic circuits and to avoid a reluctance to use AI models which seemingly only demonstrate a benefit on the optimal patient; and the seamless integration of AI models into diagnostic workflows, which requires careful consideration of ethical and legal implications, as well as the need for rigorous validation studies to ensure the safety and efficacy of AI applications.

However, it remains to be seen how fast and in what manner promising techniques like 6D fetal pose estimation, slice-to-volume registration tools, and the real-time recognition of normal and abnormal CNS anatomy, to name a few, will be integrated into clinical practice and medical education, alongside the continued advancement of the current, already commercialized AI frameworks.

## Data Availability

The authors are willing to provide additional information about their research. For more information, please contact the corresponding author.

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
