# Peer review of "Advancements in Artificial Intelligence for Fetal Neurosonography: A Comprehensive Review"

_jcm, 2024, doi:10.3390/jcm13185626_

Round 1

Reviewer 1 Report

Comments and Suggestions for Authors The search strategy was not clearly defined The type of study included was not defined so we do not know which studies they considered. This data is not even present in the summary table. The PRISMA (Preferred Reporting Items for Systematic Reviews and Meta‐Analyses) Flow chart was not reported to understand on what basis they chose which studies to include for all the reviewed studies. The limitations of their review are not reported in the conclusions.

Author Response

Response to Reviewers

Advancements in artificial intelligence for fetal neurosonography: a comprehensive review (jcm-3154847)

Reviewer 1

Comments and Suggestions for Authors

The search strategy was not clearly defined. The type of study included was not defined so we do not know which studies they considered. This data is not even present in the summary table. The PRISMA (Preferred Reporting Items for Systematic Reviews and MetaAnalyses) Flow chart was not reported to understand on what basis they chose which studies to include for all the reviewed studies.

Thank you for your valuable input. You are right. We understand the somehow confusing categorization of our review. In fact, we aimed to provide a comprehensive understanding of current AI solutions in the field of fetal neurosonography in a solely clinical context. Accordingly, we intentionally did not conduct a systematic review exclusively relying on the PRISMA criteria yet, as we are convinced that the entire width of automated, AI-based applications with clinical relevance to the fetal CNS should be included and discussed in this focused review. Following recently published systematic reviews of AI applications in ultrasound imaging in obstetrics and gynecology in general, we believe that a number of promising AI approaches specifically addressing the fetal CNS have not been sufficiently considered.

We therefore clarified the phrase within the abstract section into: ‘The paramount objective of this focusing review is to discuss the latest developments in AI applications in this field [...] (page 1, line 14) and accordingly in the first sentence of the discussion (page, lane 430).

The limitations of their review are not reported in the conclusions.

Again, thank you for your comment. We included a paragraph addressing the limitations of our paper in the discussion section.

Reviewer 2 Report

Comments and Suggestions for Authors

The authors provide an extensive overview of the application of AI in fetal neurosonography, covering its use in prenatal diagnosis, specific applications in fetal neurosonography, and potential future perspectives. Overall, this review is well-written, comprehensive, and timely. Here are some comments and suggestions for the authors to consider:

1.            The authors should include a section on the limitations and challenges of current AI applications in this field, such as the need for diverse training datasets and standardization. Additionally, it would be beneficial to discuss potential ethical considerations, including informed consent when AI is used in diagnostic processes, and the impact of AI on clinical decision-making.

2.            In the section on AI in Fetal Neurosonography, could the authors provide a comparative analysis between traditional methods and AI-assisted methods, specifically in terms of accuracy, efficiency, and clinical utility?

3.            Regarding lines 138-141, should the following sentence be removed? “3. Results

This section may be divided by subheadings. It should provide a concise and precise description of the experimental results, their interpretation, as well as the experimental conclusions that can be drawn.”

Comments on the Quality of English Language

Overall, this review is well-written.

Author Response

Response to Reviewers

Advancements in artificial intelligence for fetal neurosonography: a comprehensive review (jcm-3154847)

Reviewer 2

The authors provide an extensive overview of the application of AI in fetal neurosonography, covering its use in prenatal diagnosis, specific applications in fetal neurosonography, and potential future perspectives. Overall, this review is well-written, comprehensive, and timely. Here are some comments and suggestions for the authors to consider:

  1. The authors should include a section on the limitations and challenges of current AI applications in this field, such as the need for diverse training datasets and standardization. Additionally, it would be beneficial to discuss potential ethical considerations, including informed consent when AI is used in diagnostic processes, and the impact of AI on clinical decision-making.

We thank you for these constructive comments. Indeed, a number of recent publications provide clear support for a disclosure of information regarding AI use in aiding medical diagnosis. We have briefly and concisely outlined your suggestion in the conclusion on page 6, lines 469-479. If it is necessary for the sake of clarity, of course, we can list these lines separately under “limitations and challenges/considerations of current AI applications”.

We take a closer look at the impact of AI on clinical decision-making on page 5, lines 422-428.

We agree, diverse training data sets and standardization in image or volume acquisition are of utmost importance in the context of both efficient and responsible use of AI applications in medical imagery and healthcare in general. This can be exemplified by the now retracted manuscript by Sreelakshmy et al. on AI-assisted cerebellum segmentation. In this paper, among other methodological flaws, an algorithm was trained with ultrasound images of inadequate transcerebellar slice planes.

  1. In the section on AI in Fetal Neurosonography, could the authors provide a comparative analysis between traditional methods and AI-assisted methods, specifically in terms of accuracy, efficiency, and clinical utility?

Thank you for this further inquiry. In table 1, we have listed clinically and commercially available – i.e. practically relevant – algorithms (grey shaded rows). The 5D CNS+ software stands out in particular. We would therefore like to explain your comment in more detail using the example of the further developed, fully automated 5D CNS+ software. We recently compared it with its semi-automatic previous version. Although we compared two software versions with each other - and not a manually procedure against automatism - manual knowledge is still essential and it provides an interesting insight. The ISUOG requires the adjustment of nine planes to create a comprehensive fetal neurosonogram. In daily clinical practice, this is almost impossible to do manually and is dependent on maternal and fetal factors. The use of AI can increase accuracy, make processes more efficient and is therefore of outstanding clinical added value. The added value of AI compared to the traditional, manual acquisition of a fetal neurosonogram has already been demonstrated by other authors such as Rizzo et al., 2016 and Gembicki et al., 2023. We address this aspect on page 4, lines 180-185. If you are interested, we recommend the following white paper: Weichert J, Scharf JL. Samsung WhitePaper - AI-Enhanced 5D CNS+ - Fully Automated Tool for Standardized Plane Detection in Fetal Neurosonography (WP202404-5D CNS+TM). Published online April 2024.

  1. Regarding lines 138-141, should the following sentence be removed? “3. Results

This section may be divided by subheadings. It should provide a concise and precise description of the experimental results, their interpretation, as well as the experimental conclusions that can be drawn.”

Thank you for your kind advice. This sentence incorrectly remained during the formatting process of the Journal’s template and has now been removed.
